# Lack of parent-of-origin effects in *Nasonia* jewel wasp: A replication and extension study

**Kimberly C. Olney**[1,2], **Joshua D. Gibson**[3], **Heini M. Natri**[2], **Avery Underwood**[1,2], **Juergen Gadau**[4], **Melissa A. Wilson**[1,2,5]*

**1** School of Life Sciences, Arizona State University, Tempe, AZ, United States of America, **2** Center for Evolution and Medicine, Arizona State University, Tempe, AZ, United States of America, **3** Department of Biology, Georgia Southern University, Statesboro, GA, United States of America, **4** Institut fuer Evolution and Biodiversity, University of Muenster, Muenster, Germany, **5** Center for Mechanisms of Evolution, The Biodesign Institute, Arizona State University, Tempe, AZ, United States of America

* mwilsons@asu.edu

**Data Availability Statement:** Scripts and gene lists used to analyze these data are publicly available on GitHub, https://github.com/SexChrLab/Nasonia.

## Abstract

In diploid cells, the paternal and maternal alleles are, on average, equally expressed. There are exceptions from this: a small number of genes express the maternal or paternal allele copy exclusively. This phenomenon, known as genomic imprinting, is common among eutherian mammals and some plant species; however, genomic imprinting in species with haplodiploid sex determination is not well characterized. Previous work reported no parent-of-origin effects in the hybrids of closely related haplodiploid *Nasonia vitripennis* and *Nasonia giraulti* jewel wasps, suggesting a lack of epigenetic reprogramming during embryogenesis in these species. Here, we replicate the gene expression dataset and observations using different individuals and sequencing technology, as well as reproduce these findings using the previously published RNA sequence data following our data analysis strategy. The major difference from the previous dataset is that they used an introgression strain as one of the parents and we found several loci that resisted introgression in that strain. Our results from both datasets demonstrate a species-of-origin effect, rather than a parent-of-origin effect. We present a reproducible workflow that others may use for replicating the results. Overall, we reproduced the original report of no parent-of-origin effects in the haplodiploid *Nasonia* using the original data with our new processing and analysis pipeline and replicated these results with our newly generated data.

## Introduction

Parent-of-origin effects occur when there is a biased expression (or completely monoallelic expression) of alleles inherited from the two parents [1, 2]. Monoallelic gene expression in the offspring is hypothesized to be primarily the result of genetic conflict between parents over resource allocation in the offspring [3, 4]. In mammals, the mechanism of these parent-of-origin effects occurs via inherited methylation of one allele [1, 5]. In insects, the relationship between methylation of genomic DNA and the expression of the gene that it encodes is not as well characterized but studies of social insects showed that there is a positive correlation of DNA methylation of gene bodies and gene expression [6].

Newly generated data are available at:
PRJNA613065.

**Funding:** ARCS Spetzler Scholar additionally
supported KCO. https://www.arcsfoundation.org/
national-homepage. HMN was supported by an
ASU Center for Evolution and Medicine
postdoctoral fellowship. https://evmed.asu.edu/
MAW was supported by the National Institute of
General Medical Sciences (NIGMS) of the National
Institutes of Health (NIH) grant R35GM124827.
https://www.nih.gov/. JDG was supported by the
Division of Integrative Organismal Systems (IOS)
of the National Science Foundation (NSF) grant
1145509 and by research funds provided by
Georgia Southern University. https://www.nsf.gov/.
JG was in part supported by a grant from the
German Research Foundation (DFG) to N.S.
(281125614/GRK2220, Project B7). https://www.
dfg.de/en/. The funders had no role in study
design, data collection and analysis, decision to
publish, or preparation of the manuscript.

**Competing interests:** The authors have declared
that no competing interests exist.

**Abbreviations:** ASE, Allele-specific expression; DE,
Differential expression; DEGs, Differentially
expressed genes; FDR, False discovery rate;
$\log_2$FC, $\log_2$ fold-change; logCPM, log counts per
million; GG, *Nasonia giraulti* maternal and paternal;
VV, *Nasonia vitripennis* maternal and paternal;
$F_1$GV, *N. giraulti* maternal, *N. vitripennis paternal*
(GV); $F_1$VG, *N. vitripennis* maternal, *N. giraulti*
*paternal* (VG); ND, No difference; RNAseq, RNA
sequence; SD, Standard deviation; VCF, Variant call
format.

Honey bees have been a focal group for investigation of parent-of-origin effects in insects due to differences in the kinship between queens, males, and workers [7, 8]. Multiple mating by queens results in low paternal relatedness between workers and should lead to intragenomic conflict over worker reproduction (laying unfertilized eggs to produce males), and ultimately should favor the biased expression of paternal alleles that promote worker reproduction [9]. Utilizing a cross between European (*Apis mellifera ligustica*) and Africanized honey bees, Galbraith et al. 2016 identified genes exhibiting a pattern of biased paternal allele overexpression in worker reproductive tissue from colonies that were queenless and broodless, a colony condition that promotes worker reproduction [9]. Smith et al. 2020 found a similar pattern of paternal allele overexpression in diploid (worker-destined) eggs in a cross between two African subspecies, *A.m. scutellate* and *A.m. capensis* [10]. In reciprocal crosses of European (*A.m. ligustica* and *A.m. carnica*) and Africanized honey bees reared in colonies containing both brood and a queen, Kocher et al. 2015 instead found parent-of-origin effects in gene expression that were largely overexpressing the maternal allele in both directions of the cross [11]. Recent work by Marshall et al. 2020 has also identified parent-of-origin effects in the bumblebee, *Bombus terrestris* [12]. These studies provide evidence for parent-of-origin effects in honey bees and bumblebees, both eusocial Hymenoptera. The Kocher et al. 2016 honey bee dataset also exhibited asymmetric maternal allelic bias in which the paternal allele was silenced, but only in hybrids with Africanized fathers [13]. This set of biased genes was enriched for mitochondrial-localizing proteins and is overrepresented in loci associated with aggressive behavior in previous studies [14, 15]. Interestingly, these same crosses exhibit high aggression in the direction of the cross with the Africanized father but not in the reciprocal cross [16], and aggression and brain oxidative metabolic rate appears to be linked in honey bees [17]. This study points toward a potential role of allelic bias and nuclear-mitochondrial genetic interactions in wide crosses of honey bees.

The parasitoid wasp genus *Nasonia* has emerged as an excellent model for studying genomic imprinting in Hymenoptera. Like honey bees and all Hymenoptera, *Nasonia* has a haplodiploid sex-determination system in which females are diploid, developing from fertilized eggs, and males are haploid, developing from unfertilized eggs. However, it serves as a strong contrast to studying parent-of-origin effects in the eusocial Hymenoptera as *Nasonia* is solitary and singly-mated, which should result in less genomic conflict and therefore less selective pressure for genomic imprinting based on kinship. By studying allelic expression biases in this system, we can better assess genomic imprinting in the absence of kin selection and the potential contribution of nuclear-mitochondrial interactions to biased allelic expression. *Nasonia* is well-suited for these kinds of studies as two closely related species of *Nasonia*—*N. vitripennis* and *N. giraulti*—that diverged ~1 million years ago (Mya) and show a synonymous coding divergence of ~3% [18], can still produce viable and fertile offspring [19]. Highly inbred laboratory populations of *N. vitripennis* and *N. giraulti* with reduced polymorphism provide an ideal system for identifying parent-of-origin effects in hybrid offspring [20]. However, the species do show genetic variation and incompatibilities, such that recombinant F2 males (from unfertilized eggs of F1 hybrid females) suffer asymmetric hybrid breakdown in which 50% to 80% of the offspring die during development [19]. The mortality is dependent on the direction of the cross and those with *N. giraulti* maternity (cytoplasm) have the highest level of mortality. Nuclear-mitochondrial incompatibilities have been implicated in this and candidate loci have been identified [21–23]. Despite this high level of mortality in F2 males, there is no obvious difference in mortality of the F1 mothers of these males and non-hybrid females, further highlighting this as an excellent system in which to test the potential role of allelic expression bias in mitigating hybrid dysfunction.

Wang et al. 2016 used genome-wide DNA methylation and transcriptome-wide gene expression data from 11 individuals to test whether differences in DNA methylation drive the

differences in gene expression between *N. vitripennis* and *N. giraulti*, and whether there are any parent-of-origin effects (parental imprinting and allele-specific expression) [20]. They used reciprocal crosses of these two species and found no parent-of-origin effects, suggesting a lack of genomic imprinting. Unlike the work in honey bees and bumblebees, however, there have not been multiple independent investigations of evidence for parent-of-origin effects in *Nasonia*.

Reproducibility is a major concern in science, particularly for the biological and medical sciences [24, 25]. To replicate is to make an exact copy. To reproduce is to make something similar to something else. Reports have shown that significant factors contributing to irreproducible research include selective reporting, unavailable code and methods, low statistical power, poor experimental design, and raw data not available from the original lab [24, 26, 27]. In RNAseq experiments, raw counts are transformed into gene or isoform counts, which requires an *in silico* bioinformatics pipeline [28]. These pipelines are modular and parameterized according to the experimental setup [28]. The choice of software, parameters used, and biological references can alter the results. In RNAseq, filters can also improve the robustness of differential expression calls and consistency across sites and platforms [29]. There is no, and there may never be, a defined optimal RNAseq processing pipeline from raw sequencing files to meaningful gene or isoform counts. Thus, the same data can be processed in a multitude of ways by the choice of software, parameters, and references used [28]. Given the exact same inputs, software, and parameters, one can reproduce the analysis if the authors provide this documentation and make explicit the information related to the data transformation used to the RNAseq data [28]. In the case of Wang et al. 2016, the methods and experimental design were exceptionally well documented, and the authors made available their raw data [20].

To address whether the Wang et al. 2016 findings of lack of parent-of-origin effects in *Nasonia* can be replicated and reproduced, we conducted two sets of analyses. We first downloaded the raw data from 11 individuals [20] and replicated differential expression (DE) and allele-specific expression (ASE) analyses. This allowed us to characterize species differences in gene expression, hybrid effects relative to each maternal and paternal line, and possible parent-of-origin effects using new alignment methods and software. We first downloaded the raw data from 11 individuals [20] and replicated differential expression (DE) and allele-specific expression (ASE) analyses. This allowed us to characterize species differences in gene expression, hybrid effects relative to each maternal and paternal line, and possible parent-of-origin effects using new alignment methods and software. Our alignment methods differ from the original Wang et al. 2016 in several ways. Wang et al. 2016 aligned RNAseq reads to both the *N. vitripennis* and *N. giraulti* reference genomes (v1.0) using TopHat v2.0 [30], whereas in this study we created a pseudo *N. giraulti* reference and aligned the reads using HISAT [31]. HISAT has been shown to outperform TopHat in percentage of total reads aligning correctly [32]. Additionally, it has been shown that there is variation in the genes identified to be differentially expressed depending on the choice of read aligner [33, 34]. Thus, in addition to the biological differences, we would expect different transcript abundances than what were originally reported in the Wang et al. 2016 study. Second, we reproduced the experimental setup with new individuals, generated transcriptome-wide expression levels of 12 *Nasonia* individuals (parental strains and reciprocal hybrids), named here as the Wilson data using similar, but not identical strains as the Wang et al. 2016 samples, which we named as the R16A Clark data. The Wilson data, reported here, used the standard *N. giraulti* strain (RV2Xu). The R16A Clark *N. giraulti* differs from the RV2Xu strain in that it has a nuclear *N. giraulti* genome introgressed into a *N. vitripennis* cytoplasm which harbor *N. vitripennis* mitochondria. Both studies used the same highly inbred standard *N. vitripennis* strain, ASymCx. We expect that there may be some differences between the two datasets due to the strains used; as expected, we found two loci that retained some *N. vitripennis* nuclear genes but we also discovered more and

symmetric biased expression. We completed the above analyses to test for robust reproducibility in biased allele and parent-of-origin effects in *Nasonia*. In this analysis, we processed both the R16A Clark and Wilson data using the same software and thresholds, starting with the raw FASTQ files. While we detect some differences in the specific differentially expressed genes between the two datasets, our study reproduces and confirms the main conclusions of the Wang et al. 2016 study: we observe similar trends in the DE and ASE genes, and we detect no parent-of-origin effects in *Nasonia* hybrids, indicating a validation of the lack of epigenetic reprogramming during embryogenesis in this taxa [20]. We make available the bioinformatics processing and analysis pipeline used for both the R16A Clark and Wilson datasets for easily replicating the results reported here: https://github.com/SexChrLab/Nasonia. Finally, during the process of reproducing these results, we extend them to show potential interactions between the mtDNA and autosomal genome that were not apparent in the original study.

## Results

### Samples cluster by species and hybrid in R16A Clark and Wilson datasets

We used Principal Component Analysis (PCA) of gene expression data to explore the overall structure of the two datasets, R16A Clark and Wilson. Although the reciprocal hybrids from the two datasets are slightly different **Fig 1B,** in both sets, samples from the two species (strains) form separate clusters, with the clustering of the hybrid samples between them **Fig 2A**. The first PC explains most of the gene expression variation in both datasets, with proportions of variance explained 58.17% in R16A Clark and 61.69% in the Wilson data. Further, despite differences in experimental protocols, the transcriptome-wide gene expression measurements across the different crosses and species are highly correlated between the R16A Clark and Wilson dataset, **Fig 3**. There is a difference in the mean RNAseq library size between the two datasets. The mean RNAseq library size for the R16A Clark samples is 2,501,794,109 base pairs (bp) (SD = 603,925,921) and the Wilson samples is 3,326,933,217 bp (SD = 677,004,245), **S1 Table**. The mean number of reads for the R16A Clark samples is 49,054,786 (SD = 11,841,685) and the Wilson samples is 16,634,666 (SD = 3,385,021), **S1 Table**. Additionally, the R16A Clark data was sequenced to 51-bp single-end short reads per replicate [20]; whereas the Wilson samples were sequenced to 100-bp paired-end read per replicate. Overall, we observe that most of the variation in the data is explained by species and hybrids

### Species and hybrid differences in gene expression between closely related *N. vitripennis* and *N. giraulti*

We detect more differentially expressed genes (DEGs) in the Wilson dataset, particularly in the comparison involving the hybrid samples (**Fig 2B**). We called DEGs, FDR $\leq$ 0.01, and absolute $\log_2$ fold change $\geq$ 2, between the different species and crosses within both datasets (**Fig 2B** and **S1 Fig**). In the *N. vitripennis* (VV) x *N. giraulti* (GG) comparison, we identify 799 and 1,001 DEGs in the R16A Clark and Wilson datasets, respectively. We observe a 45.5% overlap of these DEGs between the datasets (**S1 Fig**). As expected, we detect fewer DEGs in the comparisons involving the hybrids (**Fig 1B**). We detect only small differences in the numbers of DEGs called in the R16A Clark and Wilson datasets when examining hybrid effects relative to each maternal line (**S1 Fig**). However, these DEGs show little overlap between the datasets, with the proportions of overlapping DEGs in VVxVG, VVxGV, GGxVG, and GGxGV comparisons being 24.1%, 16.2%, 39%, and 31.6%, respectively.

There is a notable difference in the number of DEGs called between VG and GV hybrids between the R16A Clark and Wilson datasets. The R16A Clark data used an introgression

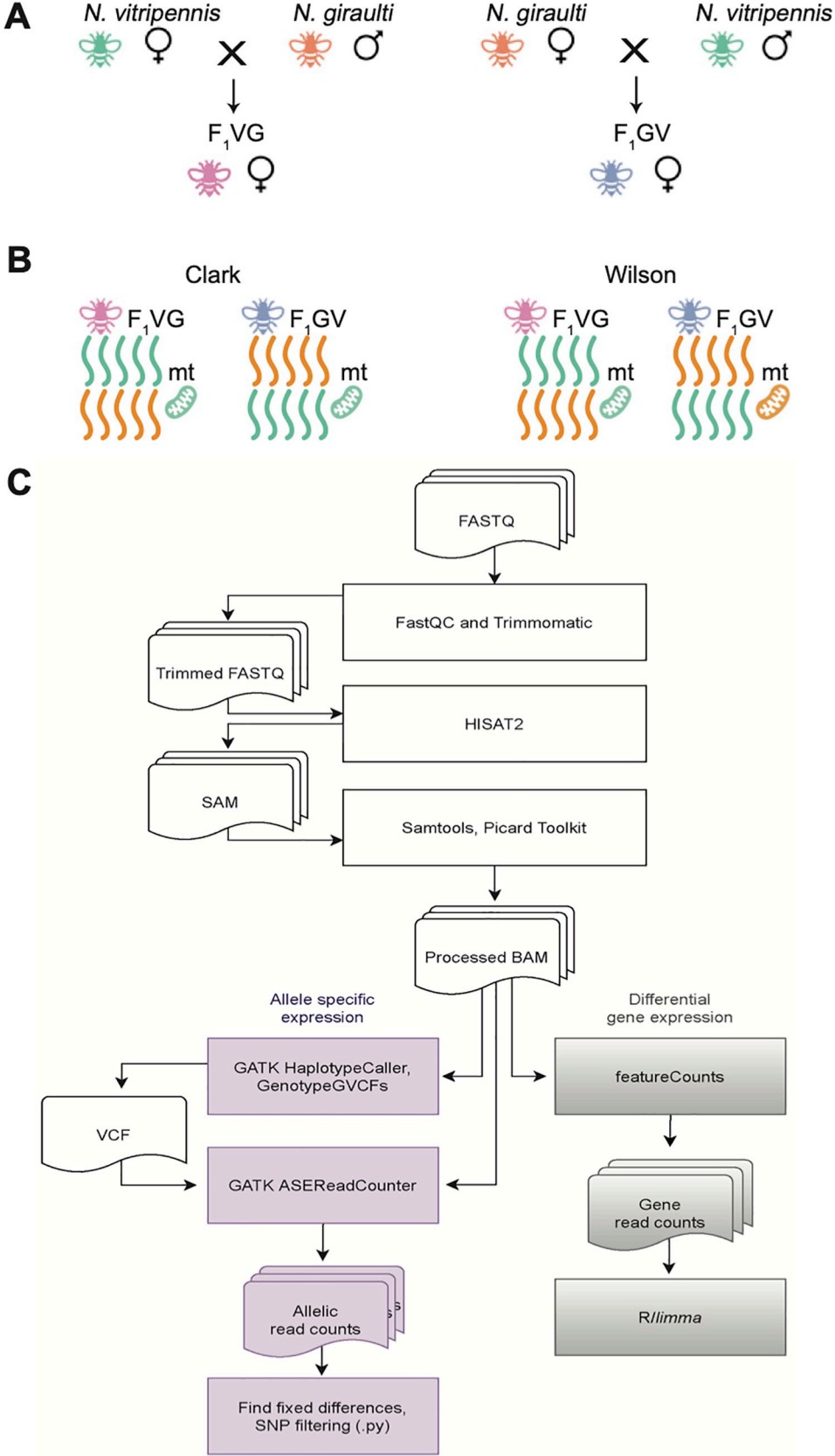

**Fig 1.**

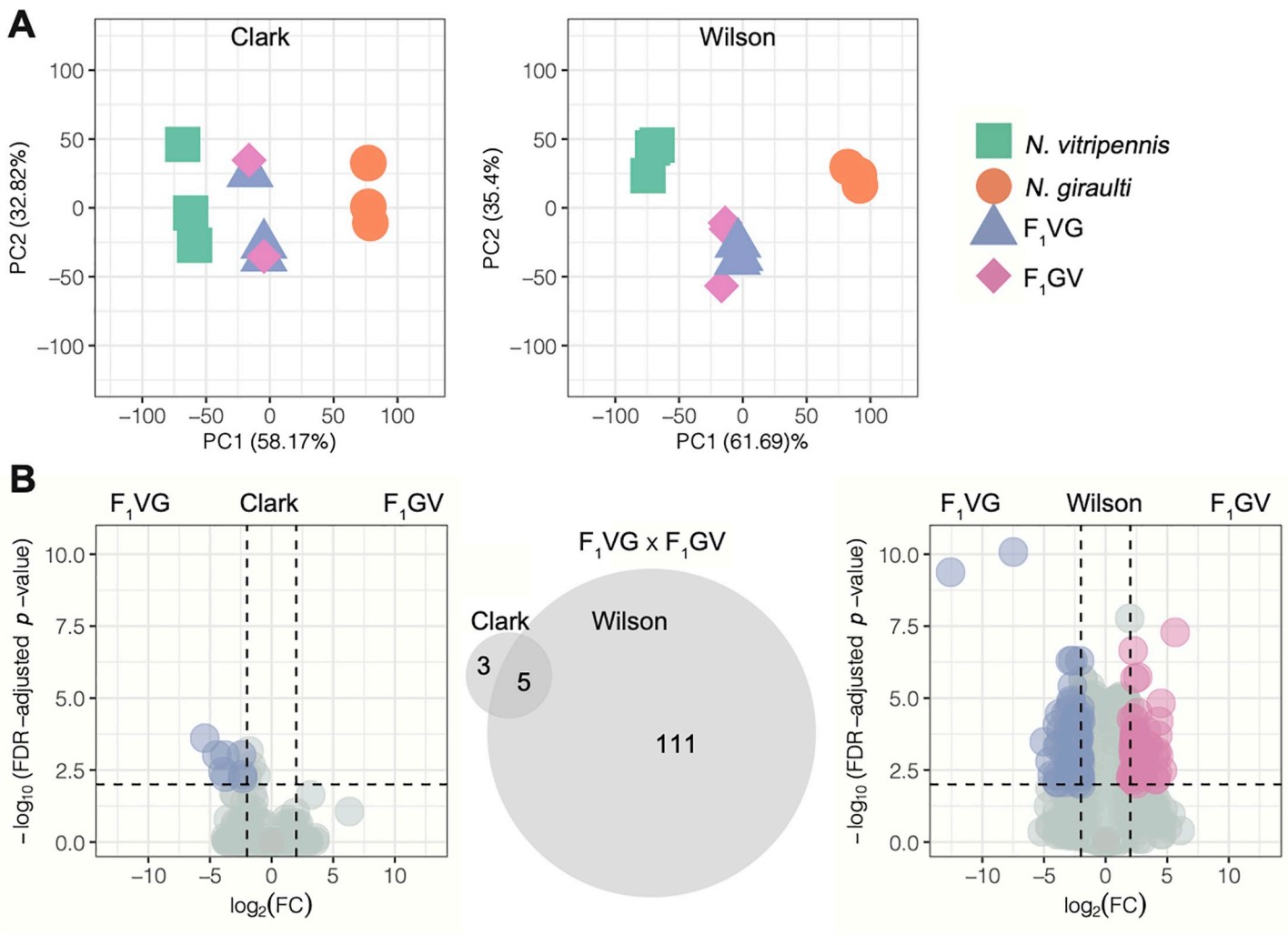

**Fig 2.**

strain of *N. giraulti*, R16A, that has a nuclear genome derived from *N. giraulti* but maintains *N. vitripennis* mitochondria, therefore the R16A Clark hybrids all have the same genetic makeup whereas the Wilson reciprocal hybrids have the same nuclear genome but different cytoplasms; yet, we do see eight genes called as differentially expressed between the VG and GV hybrids in the R16A Clark data. Three of the eight genes in the R16A Clark data (LOC116416025, LOC116416106, LOC116417553) were only called as differentially expressed between the VG and GV hybrids in the R16A Clark dataset and weren't called as differentially expressed in the Wilson dataset. The other five genes (LOC107981401, LOC100114950, LOC116415892, LOC103317241, LOC107981942) were called as differentially expressed between the VG and GV in both datasets. In the Wilson data, we called 116 DEGs, 111 of which are unique to the Wilson data set. The original Wang et al. 2016 publication did not investigate differential expression between the hybrids [20]. Here we report a new way of look-ing at the data, and despite the same genetic makeup between the hybrids in the R16A Clark data, we do observe differential expression between the hybrids, and five of those eight genes are also called as differentially expressed in the Wilson data.

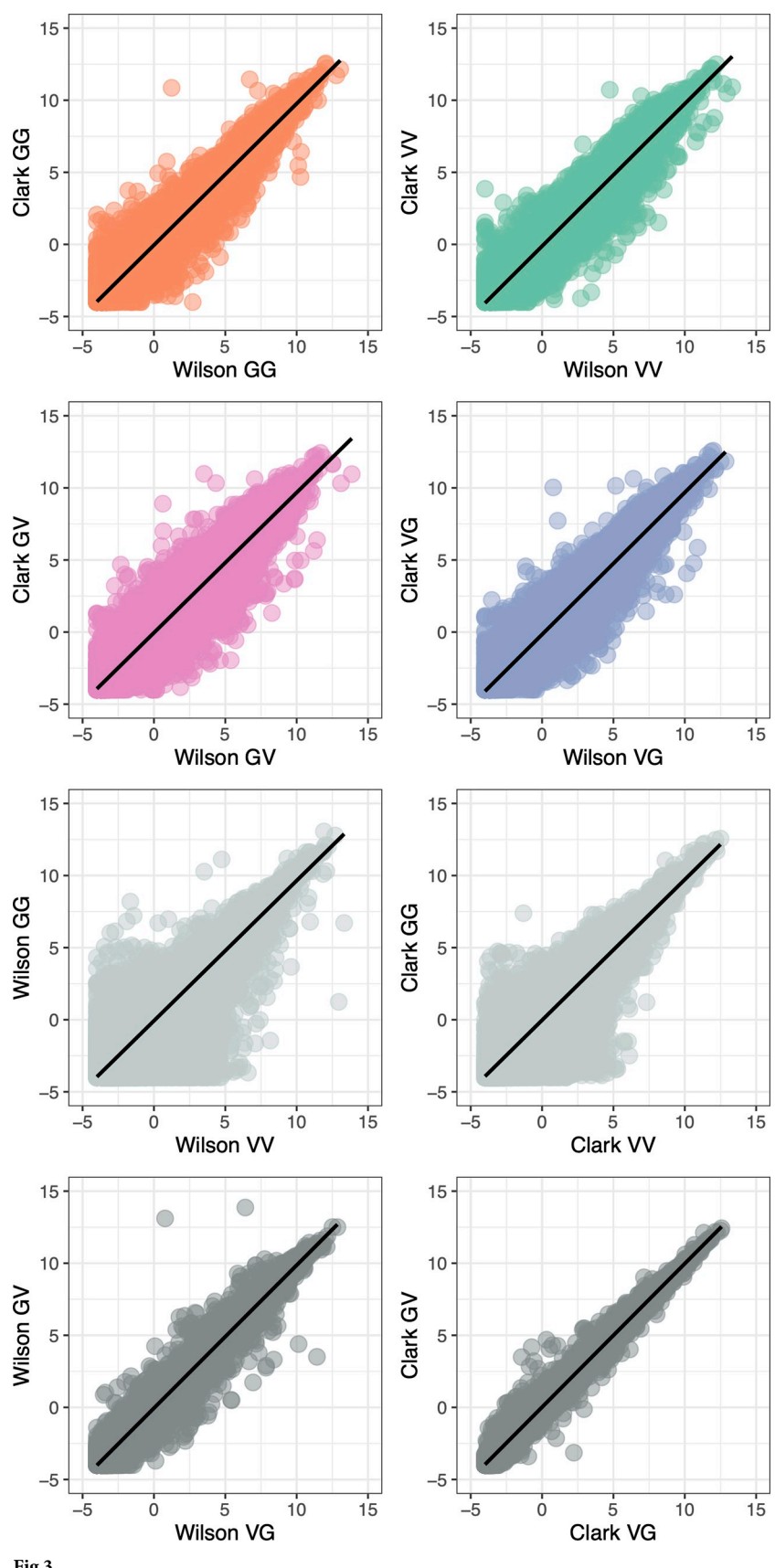

**Fig 3.**

Four (LOC107981401, LOC100114950, LOC116415892, and LOC103317241) out of the five DEGs shared between the data sets are uncharacterized proteins located on Chr 1, Chr 2, and Chr 4. To gain insight into the possible functions of these genes, we used NCBI's BLASTp excluding *Nasonia* [35, 36] to find regions of similarity between these sequences and characterized sequences. We observe several significant hits to different insects including *Drosophila* suggesting that these proteins have at least some conservation in insects over > 300 million years. The fifth shared DEG, LOC107981942, located on chromosome 1, is annotated as a zinc finger BED domain-containing protein 1. An NCBI Conserved Domain Search (https://www.ncbi.nlm.nih.gov/Structure/cdd/wrpsb.cgi) using these protein sequences uncovered no significant hits with LOC100114950, LOC116415892, and LOC103317241. However, LOC107981401 and LOC107981942 show significant hits for transposase domain superfamilies cl24015 and cl04853, respectively. The role of these proteins in *Nasonia* remains unclear.

## Lack of parent-of-origin effects in *Nasonia* hybrids

We used allele-specific expression (ASE) analyses to detect parent-of-origin effects—indicated by allelic bias—in *Nasonia* hybrids. The inference of genomic imprinting for each dataset was limited to those sites that meet our filtering criteria (see Methods). We find 107,206 and 115,490 sites to be fixed and different between VV and GG samples, in the R16A Clark and Wilson datasets, respectively. Limiting the analysis to only fixed and different sites, there are 6,377 and 7,164 genes with at least 2 informative Single Nucleotide Polymorphisms (SNPs) in the reciprocal hybrids in the R16A Clark data set and Wilson datasets, respectively. Using this approach, we find no evidence of genomic imprinting in whole adult female samples of *Nasonia* in the R16A Clark data (**Fig 4A**). But for the Wilson data we found two genes that show a pattern of expression consistent with genomic imprinting: CPR35 and LOC103315494. In the VG hybrid, CPR35 shows a bias towards the paternally inherited *N. giraulti* allele at an allele ratio of 65.3% and in the GV hybrid towards the paternally inherited *N. vitripennis* allele, with an allele ratio of 62% (**S2 Table**). CPR35 is a cuticular protein in the RR family member 35. The number of SNPs for CPR35 in the GV and VG hybrids is 2 and 4 respectively. The allele depth for CPR35 in the GV and VG hybrids is 48.5 and 63, respectively. Similarly, LOC103315494 shows bias towards the paternally inherited allele with allele ratios of 65.26% and 61.58% in VG and GV, respectively (**S2 Table**). The number of SNPs for LOC103315494 in the GV and VG hybrids is 7 in both hybrids. The allele depth for LOC103315494 in the GV and VG hybrids is 99.45 and 177.17, respectively. Although both imprinted genes, *CPR35* and *LOC103315494*, fall below the mean allele depth of 149.65 and 197.33 in the GV and VG hybrids respectively and average number of SNPs per gene at 19.45 and 19.72 in the GV and VG hybrids respectively, both genes are above the thresholds applied here (**S3 Table**).

We combined the allele-specific expression data from the R16A Clark and Wilson datasets to detect parent-of-origin effects. Data processing scripts are available on the GitHub page: https://github.com/SexChrLab/Nasonia. Only sites that are shared between the R16A and Wilson datasets were used for inference of genomic imprinting. We observe 5,759 genes with at least 2 informative SNPs in the reciprocal hybrids in the combined R16A Clark and Wilson dataset (**S2 Table**). Much like the observations in the R16A Clark and Wilson datasets, we find no evidence of genomic imprinting in whole adult female samples of *Nasonia* in the combined R16A Clark and Wilson dataset (**S2 Fig**). Eight genes show a difference in allelic expression between the VG and GV hybrids when we combine the Clark and Wilson datasets. All eight genes were previously called showing a difference in allelic expression between the F1 hybrids in either the R16A Clark or Wilson data set.

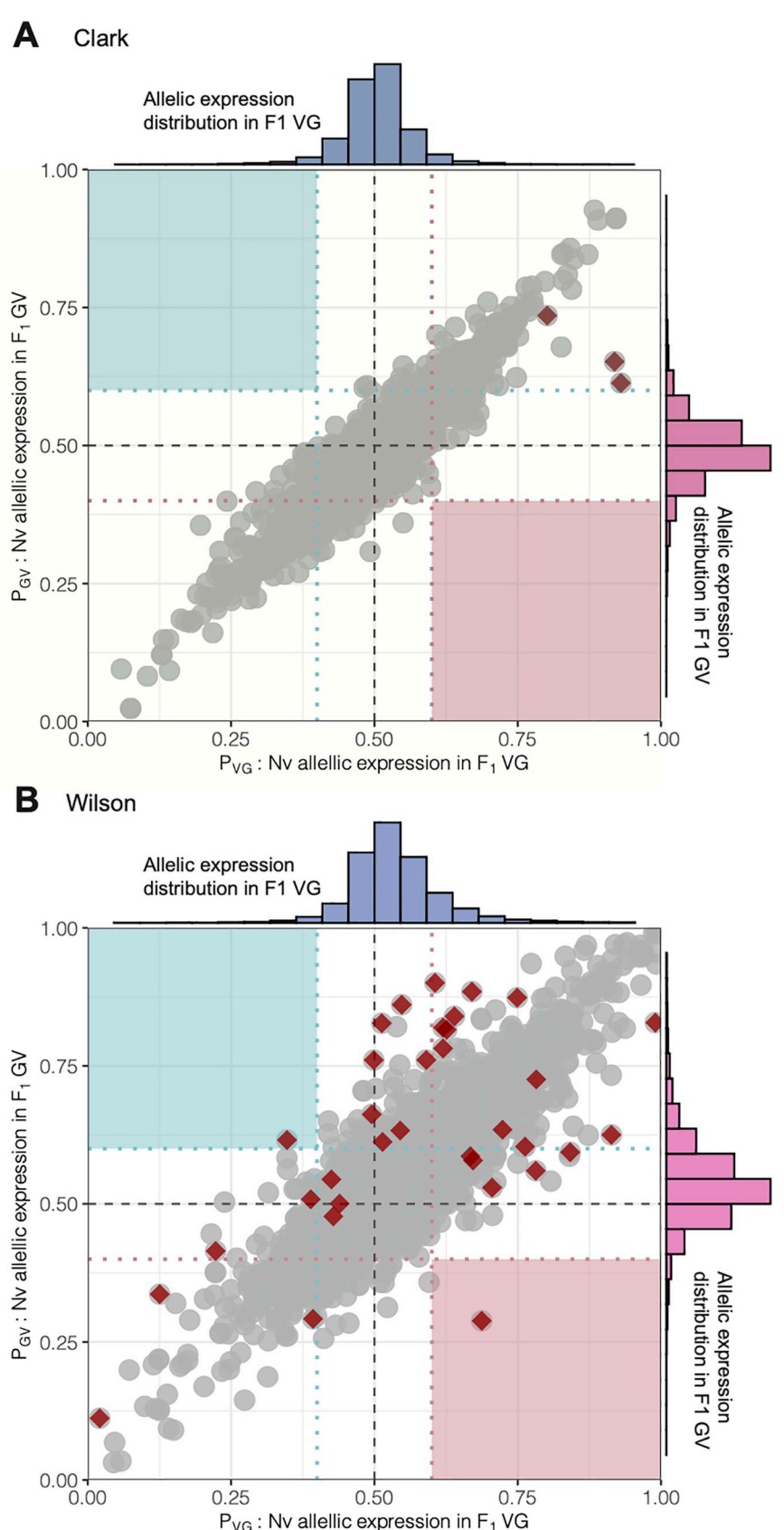

**Fig 4.**

## Allele-specific expression differences in *Nasonia* hybrids

We find three genes with higher expression of the *N. vitripennis* allele in both hybrids, in both datasets, indicative of *cis*-regulatory effects. The genes LOC100123729, LOC100123734, and LOC100113683 show consistent differences in allelic expression between VG and GV hybrids (FDR-$p \leq 0.05$) in both datasets, but the ratio of the *N. vitripennis* allele differs between the hybrids (**S2 Table**). In the R16A Clark dataset: LOC100123729 in the VG hybrids the *N. vitripennis* allele accounts for 93% of the reads, whereas in the GV hybrids this ratio is 61%. In the Wilson dataset, both hybrids showed higher expression of the *N. vitripennis* allele. In the Wilson data, the *N. vitripennis* allele ratio was 61% in VG and 90% in GV. LOC100123729 is located on chromosome 2 and encodes the protein Nasonin-3, which plays a role in inhibiting host insect melanization [37]. Also on chromosome 2 is LOC100123734, annotated as cadherin-23, which is involved in cell attachment by interacting with other proteins in the cell membrane. Both hybrids in both datasets show a higher expression for the *N. vitripennis* allele for LOC100123734. In the R16A Clark data, the ratio of the *N. vitripennis* allele in VG was 92% and in GV 65%. In the Wilson data, the VG hybrids showed less expression for the *N. vitripennis* allele than the GV hybrids, at a ratio of 64% and 84% of the reads, respectively. Finally, LOC100113683, which is located on chromosome 4, and is annotated as a general odorant-binding protein 56d also shows more expression for the *N. vitripennis* allele in both datasets and both hybrids (80.13% and 73.54% for VG and GV in R16A Clark, 78.22% and 72.57% in Wilson). Odorant binding proteins are thought to be involved in the stimulation of the odorant receptors by binding and transporting odorants which activate the olfactory signal transduction pathway [38].

## R16A strain retains *N. vitripennis* alleles

R16A is a strain produced by backcrossing an *N. vitripennis* female to an *N. giraulti* male and repeating that for 16 generations [19]. This should give a complete *N. giraulti* nuclear genome with *N. vitripennis* mitochondria. However, we identified two regions in the R16A strain that still show *N. vitripennis* alleles and named them R16A non-introgressed locus 1 and R16A non-introgressed locus 2 (**S4 Table**). Each region is identified by a single marker that retains the *N. vitripennis* allele. Locus 1 contains 44 genes and Locus 2 contains 14 genes. Both of these regions are found on Chromosome 1, and Locus 2 lies within the confidence intervals of the mortality locus for *N. vitripennis* maternity hybrids identified by Niehuis et al. 2008 [22] (i.e., F2 recombinant hybrids with a *N. vitripennis* cytoplasm showed a significant transmission ratio distortion at this region favoring the *N. vitripennis* allele). R16 A non-introgressed locus 1 harbors a mitochondrial ribosomal gene (39 S ribosomal protein 38) which is a good candidate gene for causing its retention in R16A despite intensive introgression. It would also explain the observed nuclear-cytoplasmic effect in F2 recombinant males in a vitripennis cytoplasm, despite the fact that R16A was used as a giraulti parental line in Gadau et al. 1999 [21]. Gadau et al. interestingly also mapped one of the nuclear-cytoplasmic incompatibility loci to chromosome 1 (called LG1 in the manuscript) [21]. Mutations in mitochondrial ribosomal proteins in humans have severe effects [39].

## Expression of genes in regions associated with hybrid mortality or nuclear-mitochondrial incompatibility

We compared the location of genes with either significant differential gene expression or significant differences in allele-specific expression between VG and GV hybrids to the location of previously identified mortality-associated loci. Three of the five genes that were called as

differentially expressed between VG and GV hybrids in both the R16A Clark and Wilson data sets (**S5 Table**) are located within mortality-associated loci. LOC103317241 is located within a locus on Chr 2 that is associated with mortality in VG hybrids, and LOC107981401 and LOC100114950 are within a locus on Chr 4 that is associated with mortality in GV hybrids. Moreover, two of the three genes showing consistent allele-specific expression in the two data sets are located near one another in the mortality-associated locus on Chr. 2 (LOC100123729 and LOC100123734). None of the genes that are differentially expressed or that exhibit allele-specific expression are located within the 2 loci that retain the *N. vitripennis* genotype in the R16A Clark strain, nor did we find any overlap of these gene sets with either the oxidative phosphorylation or the mitochondrial ribosomal proteins.

## Discussion

We successfully replicate the findings from Wang et al. 2016, showing a lack of parent-of-origin effects in *Nasonia* transcriptomes [20]. This replication occurs independently in a different laboratory, with different *Nasonia* individuals derived from a slightly different cross, different bioinformatic pipelines, and sequencing technology. Taken together, our results from both the reanalyzed R16A Clark and Wilson datasets demonstrate a species-of-origin effect but little to no parent-of-origin effect within *Nasonia* F1 female hybrids, which may have explained the lack of mortality in the F1 females relative to the F2 recombinant hybrid males. We did observe two genes that indicated a parent-of-origin effect in the Wilson dataset presented here, CPR35 and LOC103315494. Both CPR35 and LOC103315494 genes have less than the average number of mean SNPs within a gene at 4 and 7 SNPs respectively with the mean number of SNPs at 19.7. Additionally, neither hybrids for either gene show a strong bias towards the paternally inherited allele. In the VG hybrid, CPR35 shows a bias towards the paternally inherited *N. giraulti* allele at an allele ratio of 65.3% and in the GV hybrid towards the paternally inherited *N. vitripennis* allele, with an allele ratio of 62% (**S2 Table**)**.** Similarly, LOC103315494 shows bias towards the paternally inherited allele with allele ratios of 65.26% and 61.58% in VG and GV, respectively (**S2 Table**). Therefore, although both genes passed our thresholds and show a significant bias after correcting for multiple testing, adjusted p-value < 0.05; we feel that further investigation is needed to determine if the *Nasonia* species show parent-of-origin effects. We combined the two data sets to determine if this would provide a more powerful test of parent-of-origin effects, but this did not change the main results, a lack of parent-of-origin effects in *Nasonia* F1 hybrids. Given the differences in the *N. giraulti* strains in the two data sets and our finding that R16A harbors regions that are resistant to introgression, we feel it is most appropriate to analyze each data set independently. Other observed differences between the R16A Clark and Wilson dataset include the larger number of differentially expressed genes between the two parental species in our study relative to Wang et al 2016 [20] (1001 vs 799), which is most likely the result of using a standard *N. giraulti* strain (RV2Xu) rather than an introgression strain (R16A) where the nuclear genome of *N. giraulti* was introgressed into a *N. vitripennis* cytoplasm. Additionally, we found genomic regions that resisted introgression in the R16A *Nasonia* strains utilized by Wang et al. 2016 [20]. Furthermore, we present a reproducible workflow for processing raw RNA sequence samples to call differential expression and allele-specific expression openly available on the GitHub page: https://github.com/SexChrLab/Nasonia.

### Differences between the R16A Clark and Wilson datasets

The primary difference between the R16A Clark cross and the Wilson cross is the *N. giraulti* strain choice **Fig 1B**. The new crosses presented here used the strain Rv2Xu, which is a pure *N. giraulti* strain that was used for sequencing the genome [18]. Wang et al. 2016 used an

introgression strain, R16A, which has a largely *N. giraulti* nuclear genome with an *N. vitripennis* cytoplasm [20]. This strain was produced by mating an *N. vitripennis* female with an *N. giraulti* male, and then repeatedly backcrossing the strain to *N. giraulti* males for a further 15 generations [19]. Hence, both sets of hybrids should be heterozygous at every nuclear locus for species specific markers (though see above for two non-introgressed regions); however, both reciprocal R16A Clark hybrids have *N. vitripennis* mitochondria while the new hybrids have their maternal species' mitochondria. This means that in addition to looking at parent-of-origin effects, our new crosses are uniquely suited to investigate allelic expression biases in the context of nuclear-mitochondrial incompatibility and hybrid dysfunction.

## Observed differences in hybrids between data sets

We observe substantially more DEGs between the hybrids, VG and GV, in the Wilson data set compared to the R16A Clark data set. The smaller number of DEGs detected in the R16A Clark data in this particular comparison is likely partially due to the one excluded $F_1$GV sample (see Materials and methods). Another likely contributing factor is the differences in one parental strain between the Wilson and R16A Clark data sets. The Wilson data presented here consist of inbred parental *N. vitripennis* (strain AsymCX) VV and *N. giraulti* (strain RV2Xu) GG lines, and reciprocal F1 crosses. This cross differs from the R16A Clark data, which used the same *N. vitripennis* strain but rather than a normal *N. giraulti* strain they used the introgression strain, R16A, that has a nuclear genome derived from *N. giraulti* and a cytoplasm/ mitochondria derived from *N. vitripennis* (see R16A section). Despite these differences, of the eight genes that are differentially expressed between the VG and GV hybrids. five are shared between both data sets. Although we were not specifically looking for this, we found that three of the five genes showing differential expression in both data sets as well as two of the three genes showing allele (species)-specific expression in both data sets are located in previously identified loci that are associated with the observed F2 recombinant male hybrid breakdown from the same crosses [21, 22]. These findings point towards an involvement of cis regulatory elements in the genetic architecture of the F2 hybrid male breakdown in *Nasonia*. The finding that, despite using different strains of wasps, we are still able to identify genes associated with these hybrid defects bolsters our confidence in further pursuing these genes in our investigation of the genetic architecture of hybrid barriers in *Nasonia*.

## The choice of reference and tools does not alter main findings

The authors of the Wang et al. 2016 paper used different computational tools for trimming and alignment than the current study [20]. Additionally, in Wang et al. 2016, the RNAseq reads were aligned to both an *N. vitripennis* and *N. giraulti* reference genome [20]; whereas here, we created a pseudo *N. giraulti* reference genome from the fixed and differentiated sites between the inbred *N. vitripennis* and *N. giraulti* parental lines. Often, different tools and statistical approaches result in different findings [34, 40]; however, despite different approaches, we observe the same pattern as what was originally reported in Wang et al. 2016 [20], a lack of parent-of-origin expression in *Nasonia*.

## A reproducible workflow for investigating genomic imprinting

Significant factors contributing to irreproducible research include selective reporting, unavailable code and methods, low statistical power, poor experimental design, and raw data not available from the original lab [24]. We replicate a robust experimental design (current study) initially presented in the Wang et al. (2016) [20] and present a new workflow for calling DE and ASE in those two independent but analog *Nasonia* datasets. Both datasets are publicly

available for download on the short read archive (SRA) PRJNA260391 and PRJNA613065, respectively. In our analyses of the Wilson data and reanalysis of the R16A data, we corroborated the original findings from Wang et al. 2016 [20]. There are no parent-of-origin effects in *Nasonia*. All dependencies for data processing are provided as a Conda environment, allowing for seamless replication. All code is openly available on GitHub https://github.com/SexChrLab/Nasonia.

## Materials and methods

### *Nasonia vitripennis and Nasonia giraulti* inbred and reciprocal F1 hybrid datasets

RNA sequence (RNAseq) samples for 4 female samples each from parental species, *N. vitripennis* (VV) and *N. giraulti* (GG), and from each reciprocal F1 cross (F$_1$VG, female hybrids with *N. vitripennis* mothers, and F$_1$GV, female hybrids with *N. giraulti* mothers), as shown in **Fig 1A**, were obtained from Wang et al. 2016 [20] from SRA PRJNA299670. We refer to the data from [20] as R16A Clark. One F$_1$GV RNAseq sample from the R16A Clark dataset (SRR2773798) was excluded due to low quality, as in the original publication [20].

The newly generated crosses consisted of 12 RNAseq samples of inbred isofemale lines of parental *N. vitripennis* (strain AsymCX) VV and *N. giraulti* (strain RV2Xu) GG lines, and reciprocal F1 crosses F$_1$VG, and F$_1$GV. (**Fig 1A**). Whole transcriptome for these samples is available on SRA PRJNA613065. This cross differs from the R16A Clark data, which used the same *N. vitripennis* strain but rather than a standard *N. giraulti* strain used an introgression strain, R16A, that has a nuclear genome derived from *N. giraulti* and a cytoplasm/mitochondria derived from *N. vitripennis* (see R16A section below) **Fig 1B**. Total RNA was extracted from a pool of four 48 hour post-eclosion adult females using a Qiagen RNeasy Plus Mini kit (Qiagen, CA). RNA-seq libraries were prepared with 2µg of total RNA using the Illumina Stranded mRNA library prep kit and were sequenced on a HiSeq2500 instrument following standard Illumina protocols. Three biological replicates were generated for each parent and hybrid, with 100-bp paired-end reads per replicate. Sample IDs, parent cross information, and SRA bioproject accession numbers for R16A Clark and Wilson datasets are listed in **S1 Table**.

### Quality control

Raw sequence data from both datasets were processed and analyzed according to the workflow presented in **Fig 1C**. The quality of the FASTQ files was assessed before and after trimming using FastQC v0.11 [41] and MultiQC v1.0 [42]. Reads were trimmed to remove bases with a quality score less than 10 for the leading and trailing stand, applying a sliding window of 4 with a minimum mean PHRED quality of 15 in the window and a minimum read length of 80 bases, and adapters were removed using Trimmomatic v0.36 [43]. Pre- and post-trimming multiQC reports for the R16A Clark and Wilson datasets are available on the GitHub page: https://github.com/SexChrLab/Nasonia.

### Variant calling

For variant calling, BAM files were preprocessed by adding read groups with Picard's AddOrReplaceReadGroups and by marking duplicates with Picard's MarkDuplicates (https://github.com/broadinstitute/picard). Variants were called using GATK [44–46] and the scatter-gather approach: Sample genotype likelihoods were called with HaplotypeCaller minimum base quality of 2. The resulting gVCFs were merged with CombineGVCFs, and joint genotyping across

all samples was carried out with GenotypeGVCFs with a minimum confidence threshold of 10.

## Pseudo *N. giraulti* reference genome assembly

To create a pseudo *N. giraulti* reference genome, fixed differences in the homozygous *N. giraulti* and *N. vitripennis* variant call file (VCF) files were identified using a custom Python script, available on the GitHub page: https://github.com/SexChrLab/Nasonia. Briefly, a site was considered to be fixed and different if it was homozygous for the *N. vitripennis* reference allele among all three of the biological VV samples and homozygous alternate among all three of the biological GG samples. Only homozygous sites were included, as the *N. giraulti* and *N. vitripennis* lines are highly inbred. The filtered sites were then used to create a pseudo *N. giraulti* reference sequence with the FastaAlternateReferenceMaker function in GATK version 3.8 (available at: http://www.broadinstitute.org/gatk/). Reference bases in the *N. vitripennis* genome were replaced with the alternate SNP base at variant positions. Following a similar protocol for comparison, we now aligned reads in each sample to the pseudo *N. giraulti* genome reference with HISAT2 version 2.1.0, and performed identical preprocessing steps prior to variant calling with GATK version 3.8 HaplotypeCaller.

## RNAseq alignment and gene expression level quantification

Trimmed sequence reads were mapped to the NCBI *N. vitripennis* reference genome (assembly accession GCF_009193385.2), as well as the pseudo *N. giraulti* reference using HISAT2 [31]. The resulting SAM sequence alignment files were converted to BAM, and coordinates were sorted and indexed with samtools 1.8 [47]. RNAseq read counts were quantified from the *N. vitripennis* as well as the custom *N. giraulti* alignments using Subread featureCounts [48] with the *N. vitripennis* gene annotation.

## Inference of differential gene expression

Differential expression (DE) analyses were carried out by linear modeling as implemented in the R package *limma* [49]. An average of the reads mapped to each gene in the *N. vitripennis* and the pseudo *N. giraulti* genome references were used in the DE analyses. Counts were filtered to remove lowly expressed genes by retaining genes with a mean FPKM $\leq 0.5$ in at least one sample group (VV, GG, VG, or GV). Normalization of expression estimates was accomplished by calculating the trimmed mean of M-values (TMM) with edgeR [50]. The voom method [51] was then employed to normalize expression intensities by generating a weight for each observation. Gene expression is then reported as log counts per million (logCPM). Gene expression correlation between datasets and between species within each dataset was assessed using Pearson's correlation of mean logCPM values of each gene. Dimensionality reduction of the filtered and normalized gene expression data was carried out using scaled and centered PCA with the *prcomp*() function of base R. Differential expression analysis with voom was carried out for each pairwise comparison between strains (VV, GG, VG, and GV) for each data set. We identified genes that exhibited significant expression differences with an adjusted *p*-value of $\leq 0.01$ and an absolute log$_2$ fold-change (log$_2$FC) $\leq 2$.

## Analysis of allele-specific expression in reciprocal F1 hybrids

Allele-specific expression (ASE) levels were obtained using GATK ASEReadCounter [45] with a minimum mapping quality of 10, minimum base quality of 2, and a minimum depth of 30. Only sites with a fixed difference between inbred VV and GG for both R16A Clark and Wilson

datasets were used for downstream analysis of allele-specific expression. Allele counts obtained from GATK ASEReadCounter were intersected with the *N. vitripennis* gene annotation file using bedtools version 2.24.0 [52]; the resulting output contained allele counts for each SNP and corresponding gene information. The F1 hybrids' allele counts with gene information was read into R and then filtered to only include genes with at least two SNPs with minimum depth of 30. We counted the number of allele-counts for the reference allele (*N. vitripennis*) and alternative (*N. giraulti*) allele at polymorphic SNP positions. We quantified the number of SNPs in each hybrid replicate that 1) showed a bias towards the allele that came from the *N. vitripennis* parent, 2) showed a bias towards the allele that came from the *N. giraulti* parent, and 3) showed no difference (ND) in an expression of its parental alleles. The significance of allelic bias was determined using Fisher's exact test. Significant genes were selected using a Benjamini-Hochberg false discovery rate FDR-adjusted *p*-value threshold of 0.05. As *Nasonia* are haplodiploid, all ASE analyses were carried out on the diploid female hybrids.

### Identifying loci associated with hybrid mortality

*Nasonia* recombinant F2 hybrid males (haploid sons of F1 female hybrids) suffer mortality during development that differs between VG and GV hybrids [19]. Niehuis et al. 2008 identified four genomic regions associated with this mortality (i.e., regions in which one parent species' alleles are underrepresented due to mortality during development); three are associated with mortality in hybrids with *N. vitripennis* maternity and one is associated with hybrids with *N. giraulti* maternity [22]. Gibson et al. 2013 later identified a second locus related to mortality in the hybrids with *N. giraulti* maternity [23]. Given that the F1 hybrid females analyzed here experience far less mortality than their haploid male offspring, we hypothesized that these diploid females may use biased allelic expression to rescue themselves from the mortality. To compare our results with these previous studies, we had to map the previous loci to the latest *Nasonia* assembly (PSR1.1, [53]). Niehuis et al. 2008 defined their candidate loci based on the genetic distance along the chromosome (centimorgans) [22]. The physical locations of the markers along the chromosomes were later identified by Niehius et al. 2010 [54]. Using the genetic distances between these markers in both the 2008 and 2010 Niehuis *et al.* studies [22, 54], we calculated the conversion ratio between the genetic distances in these two studies (**S6 Table**). We then converted those 2008 genetic distances that correspond to the 95% Confidence Intervals for these loci to the genetic distances reported by Niehuis et al. 2010 [54], which used an Illumina Goldengate Genotyping Array (Illumina Inc., San Diego, USA) to produce a more complete and much higher resolution genetic map of *Nasonia*. This array uses SNPs to genotype samples at ~1500 loci, which allowed us to identify SNP markers that closely bound the mortality loci from the 2008 study. Gibson et al. 2013 used the same genotyping array, so this conversion was unnecessary for converting the second mortality locus in *N. giraulti* maternity hybrids [23]. We used the 100bp of sequence flanking each SNP marker to perform a BLAST search of the PSR1.1 assembly and to identify their positions. We then used all of the PSR1.1 annotated genes within these loci to look for enrichment of genes showing biased expression. Mortality loci and genomic location are reported in **S4 Table**.

### Additional gene categories of interest

Previous work has identified potential classes of genes that may be involved in nuclear-mitochondrial incompatibilities in *Nasonia*, the oxidative phosphorylation genes [55] and the mitochondrial ribosomal proteins [56]. We used the annotated gene sets from these studies to test for enrichment of genes with biased allelic expression. Lists of the genes of interest and their genomic location is reported in **S4 Table**.

## Analysis of R16A strain

In order to assess whether the introgression of the *N. giraulti* nuclear genome into the R16A Clark strain is complete, we analyzed two samples of the R16A strain using the Illumina Gold-engate Genotyping Array used in Niehuis *et al.* 2010 [54]. We searched for SNP markers that retained the *N. vitripennis* allele and only considered markers that consistently identified the proper allele in both parent species controls and that were consistent across both R16A samples, leaving 1378 markers. We defined a locus as all of the sequences between the two markers that flank a marker showing the *N. vitripennis* allele (**S2 Table**). As above, we performed a BLAST search of the PSR1.1 assembly to identify the positions of these markers. We identified all genes from the PSR1.1 assembly that lie between the flanking markers and further analyzed their expression patterns.

## Supporting information

**S1 Fig. Volcano plots for differential expression and Venn diagram of DEGs between the datasets when taking the average of the counts when aligned to *N. vitripennis* and to pseudo *N. giraulti* reference genome.** Volcano plots of DEGs detected between the different comparisons involving *N. vitripennis*, *N. giraulti*, and the two reciprocal $F_1$ hybrids in the R16A Clark (left side) and Wilson (right side) datasets. Venn diagrams of the overlap of significant DEGs in each comparison is shown.
(TIFF)

**S2 Fig. Lack of parent-of-origin effects observed when combining allele-specific expression data from the R16A Clark and Wilson datasets.** Scatterplots of the expression of the *N. vitripennis* alleles in the two reciprocal hybrids, VG (x-axis) and GV (y-axis). Analysis was limited to 5,759 genes with at least 2 informative SNPs in the reciprocal hybrids in the combined R16A Clark and Wilson dataset. Genes exhibiting a significant difference in allelic bias between the hybrids (Fisher's exact test, FDR-adj. $p<0.05$) are highlighted in red. Paternally imprinted genes are expected to appear in the upper left corner (light blue box), and maternally imprinted genes in the lower right corner (light pink box). Histograms of the *N. vitripennis* allele expression are shown for VG (blue) and GV (pink).
(TIFF)

**S1 Table. Sample identifiers.** The samples for each dataset used in the project are provided here. Samples from this study are uploaded at https://www.ncbi.nlm.nih.gov/sra/PRJNA613065.
(XLSX)

**S2 Table. Allele-specific expression differences between hybrids.** The number of allele-counts for the reference allele (*N. vitripennis*) and alternative (*N. giraulti*) allele at polymorphic SNPs within a gene. Minimum of two SNPs for a gene to be included. The significance of allelic bias was determined using Fisher's exact test. Significant genes were selected using a Benjamini-Hochberg false discovery rate FDR-adjusted *p*-value threshold of 0.05.
(XLSX)

**S3 Table. Mean and median allele and gene depth for Wilson dataset.** Mean and median allele and gene depth for each GV and VG sample in the Wilson data set. Number of SNPs for all genes, *CPR35*, and *LOC103315494*.
(XLSX)

**S4 Table. Genomic location of mortality loci and gene sets of interest.** Previously reported loci associated with mortality in *Nasonia* hybrids. 95% Confidence Intervals of loci identified

in Niehuis et al. 2008 were converted to genetic distances along the chromosomes and the closest SNP markers from Niehuis et al. 2010 were identified [22, 55]. SNP markers for the locus identified in Gibson et al. 2013 were used directly [23]. The SNP marker locations in the PSR1.1 assembly were found via BLAST and all genes within the bounds of these markers are included. The two non-introgressed regions from the R16A strain are included as well as genes from two mitochondria-associated pathways, the oxidative phosphorylation pathway [56] and the mitochondrial ribosomal proteins [56].
(XLSX)

**S5 Table. Directional bias of differentially expressed genes between VG and GV in Clark and Wilson datasets.** Five genes that were called as differentially expressed between VG and GV hybrids in both the Clark and Wilson data sets.
(XLSX)

**S6 Table. Locus conversion calculations.** Calculations for converting the genetic map positions (centimorgan, cM) of mortality loci identified by Niehuis et al. 2008 to the physical chromosomal positions of the latest genome assembly (PSR1.1) [22].
(XLSX)

## Acknowledgments

The authors acknowledge Research Computing at Arizona State University for providing HPC resources that have contributed to the research results reported within this paper.

## Author Contributions

**Conceptualization:** Joshua D. Gibson, Avery Underwood, Juergen Gadau, Melissa A. Wilson.

**Formal analysis:** Kimberly C. Olney, Joshua D. Gibson, Avery Underwood.

**Funding acquisition:** Juergen Gadau.

**Investigation:** Kimberly C. Olney, Joshua D. Gibson, Heini M. Natri, Avery Underwood, Juergen Gadau.

**Methodology:** Joshua D. Gibson, Melissa A. Wilson.

**Project administration:** Melissa A. Wilson.

**Resources:** Joshua D. Gibson, Melissa A. Wilson.

**Supervision:** Kimberly C. Olney, Joshua D. Gibson, Heini M. Natri, Juergen Gadau, Melissa A. Wilson.

**Visualization:** Kimberly C. Olney, Joshua D. Gibson, Heini M. Natri.

**Writing – original draft:** Kimberly C. Olney, Joshua D. Gibson, Heini M. Natri, Avery Underwood, Juergen Gadau, Melissa A. Wilson.

**Writing – review & editing:** Kimberly C. Olney, Joshua D. Gibson, Heini M. Natri, Avery Underwood, Juergen Gadau, Melissa A. Wilson.

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
