## [Decision Letter · Decision Letter 0]

9 Apr 2021

PONE-D-21-07342

Lack of parent-of-origin effects in Nasonia jewel wasp: a replication and extension study

PLOS ONE

Dear Dr. Wilson,

Thank you for submitting your manuscript to PLOS ONE. Your study is meritorious and deserves publication. However, there are few minor details that I would like you to consider for improving your manuscript, most importantly your choice of statistical methodology.

We look forward to receiving your revised manuscript.

Kind regards,

Olav Rueppell

Academic Editor

PLOS ONE

Journal Requirements:

Reviewers' comments:

Reviewer's Responses to Questions

**Comments to the Author**

1. Is the manuscript technically sound, and do the data support the conclusions?

Reviewer #1: Yes

Reviewer #2: Yes

2. Has the statistical analysis been performed appropriately and rigorously? 

Reviewer #1: Yes

Reviewer #2: Yes

3. Have the authors made all data underlying the findings in their manuscript fully available?

Reviewer #1: Yes

Reviewer #2: Yes

4. Is the manuscript presented in an intelligible fashion and written in standard English?

Reviewer #1: Yes

Reviewer #2: Yes

5. Review Comments to the Author

Reviewer #1: This is a reanalysis of an important data set. It also takes this further with a new dataset. They confirm the initial studies results and show some important introgression effects in the original study.

Lines 78-100. We have found a similar result in bumblebees (https://pubmed.ncbi.nlm.nih.gov/33312684/)

It would be useful in the introduction to have a brief overview of the differences between Wang and your analyses. Is there any reason why you would expect a different result?

You show CPR35 and LOC103315494 to have parent of origin effects, but they are never mentioned again. Although you state they have less than mean snps etc they are above threshold. Why do you not at least explain your logic in the discussion?

Was it not practical to combine the data from clark and wilson datasets? This would seem a more powerful test. I could see it would be not great with a fisher exact test, but what about a more powerful technique like logistic regression (see our paper). The you could have fixed factors included the direction of the cross, line, and experiment (clark/wilson)

Reviewer #2: The authors test whether parental imprinting occurs in Nasonia wasps. For this they cross different species of Nasonia, perform RNAseq to identify allele-specific expression in their offspring, and detect SNP differences between parental species. They additionally apply their analysis pipeline to an older dataset. They confirm a previous result, finding no large-scale trend of parental imprinting, but find allele-biased expression for few loci, and find expression amplitude differences at several levels (as expected, and likely to some interested in speciation/hybridization but not the main focus of the study).

This is a sweet and simple study. The manuscript is well written and thorough. The analyses & results are clear. The methods appear to be sound, with appropriate description of quality control and parameter choices The data and scripts are well described - and the github repository is easy to navigate. This is a solid study that, I feel, can be published in plos one as is.

Two minor comments:

Some support for parental imprinting is also found in bumblebees - I didn't see this explicitliy metnioned in the intro & it is worth considering mentioning https://onlinelibrary.wiley.com/doi/10.1002/evl3.197

Line 187-9 bp of reads is relevant given taht you need to have sequenced polymorphic sites to detect them. However, standard expectation is to indicate numbers of reads. So please provide those numbers in addition.

6. PLOS authors have the option to publish the peer review history of their article (what does this mean?). If published, this will include your full peer review and any attached files.

Reviewer #1: **Yes: **Eamonn Mallon

Reviewer #2: **Yes: **Yannick Wurm

---

## [Author Response · Author response to Decision Letter 0]

12 May 2021

Dear Dr. Mallon and Dr. Wurm,

Thank you for your time and effort in editing our manuscript. Below, please find our detailed response to the review comments. 

Sincerely, 

Melissa A. Wilson

Attached please find our response to the reviewer comments: 

Reviewer #1: This is a reanalysis of an important data set. It also takes this further with a new dataset. They confirm the initial studies results and show some important introgression effects in the original study.

We thank the reviewer for the suggestions to revise our manuscript. We have cited and included in the introduction a discussion of parent-of-origin effects being observed in the bumblebee in addition to the reference of imprinting observed in bumblebees. We have added to the introduction an overview of the differences between the Wang et al. 2016 analysis and what we report here. We have added to the discussion the two genes in the Wilson dataset that did show evidence of parent-of-origin effects: 

Lines 92 - 97: Recent work by Marshall et al. 2020 has also identified parent-of-origin effects in the bumblebee, Bombus terrestris [12]. These studies provide evidence for parent‐of‐origin effects in the honey bees and bumblebees, both eusocial Hymenoptera.

Lines 131 - 132: Unlike the work in honey bees and bumblebees, however, there have not been multiple independent investigations of evidence for parent-of-origin effects in Nasonia. 

Lines: 157 - 170: We expect that there may be some differences between the two datasets due to the strains used; the two studies used different N. giraulti strains for the cross. Wang et al. 2016 used R16A an introgressed N. giraulti strain that has a N. vitripennis mitochondria whereas we used Rv2x, which has a N. giraulti mitochondria. As expected, we found two loci that retained some N. vitripennis nuclear genes but we also discovered more and symmetric biased expression. Further, our alignment methods differ from the original Wang et al. 2016 in several ways. Wang et al. 2016 aligned RNAseq reads to both the r N. vitripennis and N. giraulti reference genomes (v1.0) using TopHat v2.0[30], whereas in this study we created a pseudo N. giraulti reference and aligned the reads using HISAT [31]. HISAT has been shown to outperform TopHat in percentage of total reads aligning correctly [32]. Additionally, it has been shown that there is variation in the genes identified to be differentially expressed depending on the choice of read aligner [33,34]. Thus, in addition to the biological differences, we would expect different transcript abundances than what were originally reported in the Wang et al. 2016 study.

Lines: 372 - 392: However, we did observe two genes that indicated a parent-of-origin effect in the Wilson dataset presented here, CPR35 and LOC103315494. Both CPR35 and LOC103315494 genes both have less than the average number of mean snps within a gene at 4 and 7 snps respectively with the mean number of snps at 19.7. Additionally, neither hybrids for either gene show a strong bias towards the paternally inherited allele. In the VG hybrid, CPR35 shows a bias towards the paternally inherited N. giraulti allele at an allele ratio of 65.3% and in the GV hybrid towards the paternally inherited N. vitripennis allele, with an allele ratio of 62% (S2 Table). Similarly, LOC103315494 shows bias towards the paternally inherited allele with allele ratios of 65.26% and 61.58% in VG and GV, respectively (S2 Table). Therefore, although both genes passed our thresholds and show a significant bias after correcting for multiple testing, adjusted p-value < 0.05; we feel that further investigation is needed to determine if the Nasonia species show parent-of-origin effects. We combined the two data sets to determine if this would provide a more powerful test of parent-of-origin effects, but this did not change the main results, a lack of parent-of-origin effects in Nasonia F1 hybrids. Given the differences in the N. giraulti strains in the two data sets and our finding that R16A harbors regions that are resistant to introgression, we feel it is most appropriate to analyze each data set independently. Other observed differences between the R16A Clark and Wilson dataset include the larger number of differentially expressed genes between the two parental species in our study relative to the Wang et al (2016) [20] (1001 vs 799), which is most likely the result of using a standard N. giraulti strain (RV2Xu) rather than an introgression strain (R16A) where the nuclear genome of N. giraulti was introgressed into a N. vitripennis cytoplasm.

Lines 78-100. We have found a similar result in bumblebees (https://pubmed.ncbi.nlm.nih.gov/33312684/)

We have added the following: 

Lines 92 - 97: Recent work by Marshall et al. 2020 has also identified parent-of-origin effects in the bumblebee, Bombus terrestris [12]. These studies provide evidence for parent‐of‐origin effects in the honey bees and bumblebees, both eusocial Hymenoptera.

It would be useful in the introduction to have a brief overview of the differences between Wang and your analyses. Is there any reason why you would expect a different result?

Thank you for pointing this out. We do expect some small differences. We have added the following text to the manuscript: 

Lines: 157 - 170: We expect that there may be some differences between the two datasets due to the strains used; the two studies used different N. giraulti strains for the cross. Wang et al. 2016 used R16A an introgressed N. giraulti strain that has a N. vitripennis mitochondria whereas we used Rv2x, which has a N. giraulti mitochondria. As expected, we found two loci that retained some N. vitripennis nuclear genes but we also discovered more and symmetric biased expression. Further, our alignment methods differ from the original Wang et al. 2016 in several ways. Wang et al. 2016 aligned RNAseq reads to both the r N. vitripennis and N. giraulti reference genomes (v1.0) using TopHat v2.0[30], whereas in this study we created a pseudo N. giraulti reference and aligned the reads using HISAT [31]. HISAT has been shown to outperform TopHat in percentage of total reads aligning correctly [32]. Additionally, it has been shown that there is variation in the genes identified to be differentially expressed depending on the choice of read aligner [33,34]. Thus, in addition to the biological differences, we would expect different transcript abundances than what were originally reported in the Wang et al. 2016 study.

You show CPR35 and LOC103315494 to have parent of origin effects, but they are never mentioned again. Although you state they have less than mean snps etc they are above threshold. Why do you not at least explain your logic in the discussion?

We have added the following text: 

Lines 280 – 289: The number of SNPs for CPR35 in the GV and VG hybrids is 2 and 4 respectively. The allele depth for CPR35 in the GV and VG hybrids is 48.5 and 63, respectively. Similarly, LOC103315494 shows bias towards the paternally inherited allele with allele ratios of 65.26% and 61.58% in VG and GV, respectively (S2 Table). The number of snps for LOC103315494 in the GV and VG hybrids is 7 in both hybrids. The allele depth for LOC103315494 in the GV and VG hybrids is 99.45 and 177.17, respectively. Although both imprinted genes, CPR35 and LOC103315494, fall below the mean allele depth of 149.65 and 197.33 in the GV and VG hybrids respectively and average number of SNPs per gene at 19.45 and 19.72 in the GV and VG hybrids respectively, both genes are above the thresholds applied here (S3 Table). 

And we have added: 

Lines: 372 - 392: However, we did observe two genes that indicated a parent-of-origin effect in the Wilson dataset presented here, CPR35 and LOC103315494. Both CPR35 and LOC103315494 genes both have less than the average number of mean SNPs within a gene at 4 and 7 SNPs respectively with the mean number of SNPs at 19.7. Additionally, neither hybrids for either gene show a strong bias towards the paternally inherited allele. In the VG hybrid, CPR35 shows a bias towards the paternally inherited N. giraulti allele at an allele ratio of 65.3% and in the GV hybrid towards the paternally inherited N. vitripennis allele, with an allele ratio of 62% (S2 Table). Similarly, LOC103315494 shows bias towards the paternally inherited allele with allele ratios of 65.26% and 61.58% in VG and GV, respectively (S2 Table). Therefore, although both genes passed our thresholds and show a significant bias after correcting for multiple testing, adjusted p-value < 0.05; we feel that further investigation is needed to determine if the Nasonia species show parent-of-origin effects. We combined the two data sets to determine if this would provide a more powerful test of parent-of-origin effects, but this did not change the main results, a lack of parent-of-origin effects in Nasonia F1 hybrids. Given the differences in the N. giraulti strains in the two data sets and our finding that R16A harbors regions that are resistant to introgression, we feel it is most appropriate to analyze each data set independently. Other observed differences between the R16A Clark and Wilson dataset include the larger number of differentially expressed genes between the two parental species in our study relative to the Wang et al (2016) [20] (1001 vs 799), which is most likely the result of using a standard N. giraulti strain (RV2Xu) rather than an introgression strain (R16A) where the nuclear genome of N. giraulti was introgressed into a N. vitripennis cytoplasm.

Was it not practical to combine the data from clark and wilson datasets? This would seem a more powerful test. I could see it would be not great with a fisher exact test, but what about a more powerful technique like logistic regression (see our paper). The you could have fixed factors included the direction of the cross, line, and experiment (clark/wilson)

We conducted a combined analysis, merging the two datasets. 

When combining the datasets, we still do not observe a parent-of-origin effect in these Nasonia hybrids. 

There are 8 genes (FDRq < 0.05) that show a difference in allelic expression between the VG and GV hybrids when we combine the Clark and Wilson datasets. All 8 genes were previously called as showing a difference in allelic expression between the F1 hybrids in either the R16A Clark or Wilson data set. 

We have added the following: 

Lines: 290 - 300: We combined the allele-specific expression data from the R16A Clark and Wilson datasets to detect parent-of-origin effects. Data processing scripts are available on the GitHub page: https://github.com/SexChrLab/Nasonia. Only sites that are shared between the R16A and Wilson datasets were used for inference of genomic imprinting. We observe 5,759 genes with at least 2 informative SNPs in the reciprocal hybrids in the combined R16A Clark and Wilson dataset (S2 Table). Much like the observations in the R16A Clark and Wilson datasets, we find no evidence of genomic imprinting in whole adult female samples of Nasonia in the combined R16A Clark and Wilson dataset (S2 Fig). Eight genes show a difference in allelic expression between the VG and GV hybrids when we combine the Clark and Wilson datasets. All eight genes were previously called showing a difference in allelic expression between the F1 hybrids in either the R16A Clark or Wilson data set. 

Lines: 612 - 620: S2 Fig. Lack of parent-of-origin effects observed when combining allele-specific expression data from the R16A Clark and Wilson datasets. Scatterplots of the expression of the N. vitripennis alleles in the two reciprocal hybrids, VG (x-axis) and GV (y-axis). Analysis was limited to 5,759 genes with at least 2 informative SNPs in the reciprocal hybrids in the combined R16A Clark and Wilson dataset. Genes exhibiting a significant difference in allelic bias between the hybrids (Fisher’s exact test, FDR-adj. p<0.05) are highlighted in red. Paternally imprinted genes are expected to appear in the upper left corner (light blue box), and maternally imprinted genes in the lower right corner (light pink box). Histograms of the N. vitripennis allele expression are shown for VG (blue) and GV (pink).

Reviewer #2: The authors test whether parental imprinting occurs in Nasonia wasps. For this they cross different species of Nasonia, perform RNAseq to identify allele-specific expression in their offspring, and detect SNP differences between parental species. They additionally apply their analysis pipeline to an older dataset. They confirm a previous result, finding no large-scale trend of parental imprinting, but find allele-biased expression for few loci, and find expression amplitude differences at several levels (as expected, and likely to some interested in speciation/hybridization but not the main focus of the study).

We thank the reviewer for the overview of the manuscript and agree that the findings may be of interest to a variety of researchers, but are not the main focus of the study. 

This is a sweet and simple study. The manuscript is well written and thorough. The analyses & results are clear. The methods appear to be sound, with appropriate description of quality control and parameter choices The data and scripts are well described - and the github repository is easy to navigate. This is a solid study that, I feel, can be published in plos one as is.

Thank you kindly for the positive review of our manuscript. 

Two minor comments:

Some support for parental imprinting is also found in bumblebees - I didn't see this explicitliy metnioned in the intro & it is worth considering mentioning https://onlinelibrary.wiley.com/doi/10.1002/evl3.197

We have added the following text to the manuscript: 

Lines 92 - 97: Recent work by Marshall et al. 2020 has also identified parent-of-origin effects in the bumblebee, Bombus terrestris [12]. These studies provide evidence for parent‐of‐origin effects in the honey bees and bumblebees, both eusocial Hymenoptera.

Line 187-9 bp of reads is relevant given that you need to have sequenced polymorphic sites to detect them. However, standard expectation is to indicate numbers of reads. So please provide those numbers in addition.

We have added the following text to the manuscript: 

Lines 201 - 204: The mean number of reads for the R16A Clark samples is 52.78 million and the Wilson samples is 16.46 million, S1 Table. Additionally, the R16A Clark data was sequenced to 51-bp single-end short reads per replicate [35]; whereas the Wilson samples were sequenced to 100-bp paired-end read per replicate.

---

## [Editor Report · Decision Letter 1]

17 May 2021

Lack of parent-of-origin effects in Nasonia jewel wasp: a replication and extension study

PONE-D-21-07342R1

Dear Dr. Wilson,

We’re pleased to inform you that your manuscript has been judged scientifically suitable for publication and will be formally accepted for publication once it meets all outstanding technical requirements.

Kind regards,

Olav Rueppell

Academic Editor

PLOS ONE
---

## [Editor Report · Acceptance letter]

28 May 2021

PONE-D-21-07342R1 

Lack of parent-of-origin effects in Nasonia jewel wasp: a replication and extension study 

Dear Dr. Wilson:

I'm pleased to inform you that your manuscript has been deemed suitable for publication in PLOS ONE. Congratulations! Your manuscript is now with our production department. 

Kind regards, 

on behalf of

Dr. Olav Rueppell 

Academic Editor

PLOS ONE